# 3D Multi-bodies: Fitting Sets of Plausible 3D Human Models to Ambiguous Image Data

**Benjamin Biggs**[*]
Department of Engineering
University of Cambridge
bjb56@cam.ac.uk

**Sébastien Ehrhardt**[*]
Visual Geometry Group
University of Oxford
hyenal@robots.ox.ac.uk

**Hanbyul Joo**
Facebook AI Research
Menlo Park
hjoo@fb.com

**Benjamin Graham**
Facebook AI Research
London
benjamingraham@fb.com

**Andrea Vedaldi**
Facebook AI Research
London
vedaldi@fb.com

**David Novotny**
Facebook AI Research
London
dnovotny@fb.com

## Abstract

We consider the problem of obtaining dense 3D reconstructions of humans from single and partially occluded views. In such cases, the visual evidence is usually insufficient to identify a 3D reconstruction uniquely, so we aim at recovering several plausible reconstructions compatible with the input data. We suggest that ambiguities can be modelled more effectively by parametrizing the possible body shapes and poses via a suitable 3D model, such as SMPL for humans. We propose to learn a multi-hypothesis neural network regressor using a best-of-M loss, where each of the M hypotheses is constrained to lie on a manifold of plausible human poses by means of a generative model. We show that our method outperforms alternative approaches in ambiguous pose recovery on standard benchmarks for 3D humans, and in heavily occluded versions of these benchmarks.

## 1 Introduction

We are interested in reconstructing 3D human pose from the observation of single 2D images. As humans, we have no problem in predicting, at least approximately, the 3D structure of most scenes, including the pose and shape of other people, even from a single view. However, 2D images notoriously [9] do not contain sufficient geometric information to allow recovery of the third dimension. Hence, single-view reconstruction is only possible in a probabilistic sense and the goal is to make the posterior distribution as sharp as possible, by learning a strong prior on the space of possible solutions.

Recent progress in single-view 3D pose reconstruction has been impressive. Methods such as HMR [17], GraphCMR [20] and SPIN [19] formulate this task as learning a deep neural network that maps 2D images to the parameters of a 3D model of the human body, usually SMPL [26]. These methods work well in general, but not always (fig. 2). Their main weakness is processing *heavily occluded images* of the object. When a large part of the object is missing, say the lower body of a sitting human, they output reconstructions that are often implausible. Since they can produce only one hypothesis as output, they very likely learn to approximate the mean of the posterior distribution, which may not correspond to any plausible pose. Unfortunately, this failure modality is rather common in applications due to scene clutter and crowds.

In this paper, we propose a solution to this issue. Specifically, we consider the challenge of recovering 3D mesh reconstructions of complex articulated objects such as humans from highly ambiguous

---

[*]work completed during internship at Facebook AI Research

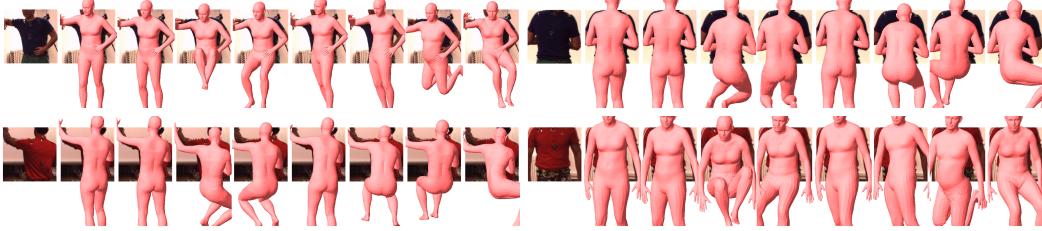

Figure 1: **Human mesh recovery in an ambiguous setting.** We propose a novel method that, given an occluded input image of a person, outputs the set of meshes which constitute plausible human bodies that are consistent with the partial view. The ambiguous poses are predicted using a novel $n$-quantized-best-of-$M$ method.

image data, often containing significant occlusions of the object. Clearly, it is generally impossible to reconstruct the object uniquely if too much evidence is missing; however, we can still predict a *set* containing all possible reconstructions (see fig. 1), making this set as small as possible. While ambiguous pose reconstruction has been previously investigated, as far as we know, this is the first paper that looks specifically at a deep learning approach for ambiguous reconstructions of the *full human mesh*.

Our primary contribution is to introduce a principled multi-hypothesis framework to model the ambiguities in monocular pose recovery. In the literature, such multiple-hypotheses networks are often trained with a so-called *best-of-$M$* loss — namely, during training, the loss is incurred only by the best of the $M$ hypothesis, back-propagating gradients from that alone [12]. In this work we opt for the *best-of-$M$* approach since it has been show to outperform alternatives (such as variational auto-encoders or mixture density networks) in tasks that are similar to our 3D human pose recovery, and which have constrained output spaces [34].

A major drawback of the *best-of-$M$* approach is that it only guarantees that *one* of the hypotheses lies close to the correct solution; however, it says nothing about the plausibility, or lack thereof, of the *other* $M-1$ hypotheses, which can be arbitrarily 'bad'.[2] Not only does this mean that most of the hypotheses may be uninformative, but in an application we are also unable to tell *which* hypothesis should be used, and we might very well pick a 'bad' one. This has also a detrimental effect during learning because it makes gradients sparse as prediction errors are back-propagated only through one of the $M$ hypotheses for each training image.

In order to address these issues, our first contribution is a *hypothesis reprojection loss* that forces each member of the multi-hypothesis set to correctly reproject to 2D image keypoint annotations. The main benefit is to constrain the *whole* predicted set of meshes to be consistent with the observed image, not just the best hypothesis, also addressing gradient sparsity.

Next, we observe that another drawback of the best-of-$M$ pipelines is to be tied to a particular value of $M$, whereas in applications we are often interested in tuning the number of hypothesis considered. Furthermore, minimizing the reprojection loss makes hypotheses geometrically consistent with the observation, but not necessarily likely. Our second contribution is thus to improve the flexibility of best-of-$M$ models by allowing them to output any smaller number

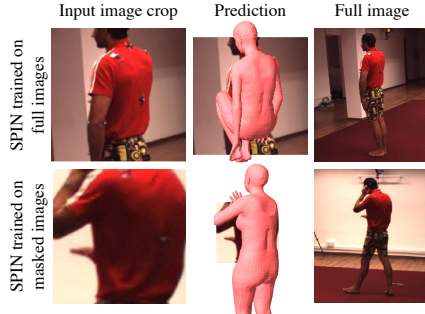

Figure 2: **Top**: Pretrained SPIN model tested on an ambiguous example, **Bottom**: SPIN model after fine-tuning to ambiguous examples. Note the network tends to regress to the mean over plausible poses, shown by predicting the missing legs vertically downward — arguably the average position over the training dataset.

$n < M$ of hypotheses while at the same time making these hypotheses *more representative of likely poses*. The new method, which we call $n$-quantized-best-of-$M$, does so by quantizing the best-of-$M$ model to output weighed by a *explicit pose prior*, learned by means of normalizing flows.

To summarise, our key contributions are as follows. First, we deal with the challenge of 3D mesh reconstruction for articulated objects such as humans in *ambiguous* scenarios. Second, we introduce a *n-quantized-best-of-$M$* mechanism to allow best-of-$M$ models to generate an arbitrary number of $n < M$ predictions. Third, we introduce a mode-wise re-projection loss for multi-hypothesis prediction, to ensure that predicted hypotheses are *all* consistent with the input.

Empirically, we achieve state-of-the-art monocular mesh recovery accuracy on Human36M, its more challenging version augmented with heavy occlusions, and the 3DPW datasets. Our ablation study validates each of our modelling choices, demonstrating their positive effect.

## 2   Related work

There is ample literature on recovering the pose of 3D models from images. We break this into five categories: methods that reconstruct 3D points directly, methods that reconstruct the parameters of a 3D model of the object via optimization, methods that do the latter via learning-based regression, hybrid methods and methods which deal with uncertainty in 3D human reconstruction.

**Reconstructing 3D body points without a model.**   Several papers have focused on the problem of estimating 3D body points from 2D observations [3, 29, 33, 41, 20]. Of these, Martinez et al. [27] introduced a particularly simple pipeline based on a shallow neural network. In this work, we aim at recovering the full 3D surface of a human body, rather than only lifting sparse keypoints.

**Fitting 3D models via direct optimization.**   Several methods *fit* the parameters of a 3D model such as SMPL [25] or SCAPE [3] to 2D observations using an optimization algorithm to iteratively improve the fitting quality. While early approaches such as [10, 37] required some manual intervention, the SMPLify method of Bogo et al. [5] was perhaps the first to fit SMPL to 2D keypoints fully automatically. SMPL was then extended to use silhouette, multiple views, and multiple people in [21, 13, 48]. Recent optimization methods such as [16, 32, 46] have significantly increased the scale of the models and data that can be handled.

**Fitting 3D models via learning-based regression.**   More recently, methods have focused on regressing the parameters of the 3D models directly, *in a feed-forward manner*, generally by learning a deep neural network [42, 43, 30, 31, 17]. Due to the scarcity of 3D ground truth data for humans in the wild, most of these methods train a deep regressor using a mix of datasets with 3D and 2D annotations in form of 3D MoCap markers, 2D keypoints and silhouettes. Among those, HMR of Kanazawa et al. [17] and GraphCMR of Kolotouros et al. [20] stand out as particularly effective.

**Hybrid methods.**   Other authors have also combined optimization and learning-based regression methods. In most cases, the integration is done by using a deep regressor to initialize the optimization algorithm [37, 21, 33, 31, 44]. However, recently Kolotouros et al. [19] has shown strong results by integrating the optimization loop in learning the deep neural network that performs the regression, thereby exploiting the weak cues available in 2D keypoints.

**Modelling ambiguities in 3D human reconstruction.**   Several previous papers have looked at the problem of modelling ambiguous 3D human pose reconstructions. Early work includes Sminchisescu and Triggs [39], Sidenbladh et al. [36] and Sminchisescu et al. [38].

More recently, Akhter and Black [1] learn a prior over human skeleton joint angles (but not directly a prior on the SMPL parameters) from a MoCap dataset. Li and Lee [22] use the Mixture Density Networks model of [4] to capture ambiguous 3D reconstructions of sparse human body keypoints directly in physical space. Sharma et al. [35] learn a conditional variational auto-encoder to model ambiguous reconstructions as a posterior distribution; they also propose two scoring methods to extract a single 3D reconstruction from the distribution.

Cheng et al. [7] tackle the problem of video 3D reconstruction in the presence of occlusions, and show that temporal cues can be used to disambiguate the solution. While our method is similar in the goal of correctly handling the prediction uncertainty, we differ by applying our method to predicting *full mesh* of the human body. This is arguably a more challenging scenario due to the increased complexity of the desired 3D shape.

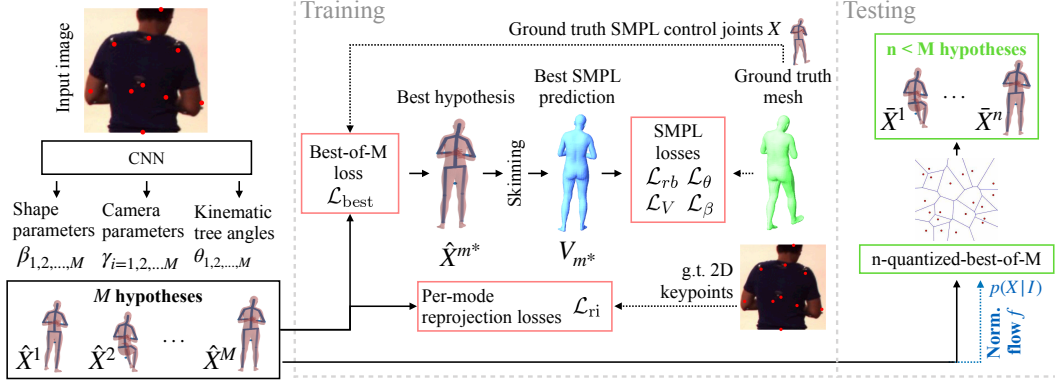

Figure 3: **Overview of our method.** Given a single image of a human, during training, our method produces multiple skeleton hypotheses $\{\hat{X}^i\}_{i=1}^M$ that enter a Best-of-$M$ loss which selects the representative $\hat{X}^{m^*}$ which most accurately matches the ground truth control joints $X$. At test time, we sample an arbitrary number of $n < M$ hypotheses by quantizing the set $\{\hat{X}^i\}$ that is assumed to be sampled from the probability distribution $p(X|I)$ modeled with normalizing flow $f$.

Finally, some recent concurrent works also consider building priors over 3D human pose using normalizing flows. Xu et al. [47] release a prior for their new GHUM/GHUML model, and Zanfir et al. [49] build a prior on SMPL joint angles to constrain their weakly-supervised network. Our method differs as we learn our prior on 3D SMPL joints.

## 3   Preliminaries

Before discussing our method, we describe the necessary background, starting from SMPL.

**SMPL.**   SMPL is a model of the human body parameterized by axis-angle rotations $\theta \in \mathbb{R}^{69}$ of 23 body joints, the shape coefficients $\beta \in \mathbb{R}^{10}$ modelling shape variations, and a global rotation $\gamma \in \mathbb{R}^3$. SMPL defines a *skinning function* $S : (\theta, \beta, \gamma) \mapsto V$ that maps the body parameters to the vertices $V \in \mathbb{R}^{6890 \times 3}$ of a 3D mesh.

**Predicting the SMPL parameters from a single image.**   Given an image $\mathbf{I}$ containing a person, the goal is to recover the SMPL parameters $(\theta, \beta, \gamma)$ that provide the best 3D reconstruction of it. Existing algorithms [18] cast this as learning a deep network $G(I) = (\theta, \beta, \gamma, t)$ that predicts the SMPL parameters as well as the translation $t \in \mathbb{R}^3$ of the perspective camera observing the person. We assume a fixed set of camera parameters. During training, the camera is used to constrain the reconstructed 3D mesh and the annotated 2D keypoints to be consistent. Since most datasets only contain annotations for a small set of keypoints ([11] is an exception), and since these keypoints do not correspond directly to any of the SMPL mesh vertices, we need a mechanism to translate between them. This mechanism is a fixed linear regressor $J : V \mapsto X$ that maps the SMPL mesh vertices $V = S(G(I))$ to the 3D locations $X = J(V) = J(S(G(I)))$ of the $K$ joints. Then, the projections $\pi_t(X)$ of the 3D joint positions into image $\mathbf{I}$ can be compared to the available 2D annotations.

**Normalizing flows.**   The idea of normalizing flows (NF) is to represent a complex distribution $p(X)$ on a random variable $X$ as a much simpler distribution $p(z)$ on a transformed version $z = f(X)$ of $X$. The transformation $f$ is learned so that $p(z)$ has a fixed shape, usually a Normal $p(z) \sim \mathcal{N}(0, 1)$. Furthermore, $f$ itself must be *invertible* and *smooth*. In this paper, we utilize a particular version of NF dubbed RealNVP [8]. A more detailed explanation of NF and RealNVP has been deferred to the supplementary.

## 4   Method

We start from a neural network architecture that implements the function $G(I) = (\theta, \beta, \gamma, t)$ described above. As shown in SPIN [19], the HMR [18] architecture attains state-of-the-art results

for this task, so we use it here. However, the resulting regressor $G(I)$, given an input image $I$, can only produce a single unique solution. In general, and in particular for cases with a high degree of reconstruction ambiguity, we are interested in predicting *set* of plausible 3D poses rather than a single one. We thus extend our model to explicitly produce a set of $M$ different hypotheses $G_m(I) = (\theta_m, \beta_m, \gamma_m, t_m)$, $m = 1, \ldots, M$. This is easily achieved by modifying the HMR's final output layer to produce a tensor $M$ times larger, effectively stacking the hypotheses. In what follows, we describe the learning scheme that drives the monocular predictor $G$ to achieve an optimal coverage of the plausible poses consistent with the input image. Our method is summarized in fig. 3.

## 4.1 Learning with multiple hypotheses

For learning the model, we assume to have a training set of $N$ images $\{I_i\}_{i=1,\ldots,N}$, each cropped around a person. Furthermore, for each training image $I_i$ we assume to know (1) the 2D location $Y_i$ of the body joints (2) their 3D location $X_i$, and (3) the ground truth SMPL fit $(\theta_i, \beta_i, \gamma_i)$. Depending on the set up, some of these quantities can be inferred from the others (e.g. we can use the function $J$ to convert the SMPL parameters to the 3D joints $X_i$ and then the camera projection to obtain $Y_i$).

**Best-of-$M$ loss.**   Given a single input image, our network predicts a set of poses, where at least one should be similar to the ground truth annotation $X_i$. This is captured by the best-of-$M$ loss [12]:

$$\mathcal{L}_{\text{best}}(J, G; m^*) = \frac{1}{N} \sum_{i=1}^{N} \left\| X_i - \hat{X}^{m_i^*}(I_i) \right\|, \quad m_i^* = \operatorname*{argmin}_{m=1,\ldots,M} \left\| X_i - \hat{X}^m(I_i) \right\|, \qquad (1)$$

where $\hat{X}^m(I_i) = J(G_m(V(I_i)))$ are the 3D joints estimated by the $m$-th SMPL predictor $G_m(I_i)$ applied to image $I_i$. In this way, only the best hypothesis is steered to match the ground truth, leaving the other hypotheses free to sample the space of ambiguous solutions. During the computation of this loss, we also extract the best index $m_i^*$ for each training example.

**Limitations of best-of-$M$.**   As noted in section 1, best-of-$M$ only guarantees that one of the $M$ hypotheses is a good solution, but says nothing about the other ones. Furthermore, in applications we are often interested in modulating the number of hypotheses generated, but the best-of-$M$ regressor $G(I)$ only produces a fixed number of output hypothesis $M$, and changing $M$ would require retraining from scratch, which is intractable.

We first address these issues by introducing a method that allows us to train a best-of-$M$ model for a large $M$ once and leverage it later to generate an arbitrary number of $n < M$ hypotheses without the need of retraining, while ensuring that these are good representatives of likely body poses.

**$n$-quantized-best-of-$M$**   Formally, given a set of $M$ predictions $\hat{\mathcal{X}}^M(I) = \{\hat{X}^1(I), ..., \hat{X}^M(I)\}$ we seek to generate a smaller $n$-sized set $\bar{\mathcal{X}}^n(I) = \{\bar{X}^1(I), ..., \bar{X}^n(I)\}$ which preserves the information contained in $\hat{\mathcal{X}}^M$. In other words, $\bar{\mathcal{X}}^n$ *optimally quantizes* $\hat{\mathcal{X}}^M$. To this end, we interpret the output of the best-of-$M$ model as a set of choices $\hat{\mathcal{X}}^M(I)$ for the possible pose. These poses are of course not all equally likely, but it is difficult to infer their probability from (1). We thus work with the following approximation. We consider the prior $p(X)$ on possible poses (defined in the next section), and set:

$$p(X|I) = p(X|\hat{\mathcal{X}}^M(I)) = \sum_{i=1}^{M} \delta(X - \hat{X}^i(I)) \frac{p(\hat{X}^i(I))}{\sum_{k=1}^{M} p(\hat{X}^k(I))}. \qquad (2)$$

This amounts to using the best-of-$M$ output as a conditioning *set* (i.e. an unweighted selection of plausible poses) and then use the prior $p(x)$ to weight the samples in this set. With the weighted samples, we can then run $K$-means [24] to further quantize the best-of-$M$ output while minimizing the quantization energy $E$:

$$E(\bar{\mathcal{X}}|\hat{\mathcal{X}}) = \mathbb{E}_{p(X|I)} \left[ \min_{\{\bar{X}^1,\ldots,\bar{X}^n\}} \|X - \bar{X}^j\|^2 \right] = \sum_{i=1}^{M} \frac{p(\hat{X}^i(I))}{\sum_{k=1}^{M} p(\hat{X}^k(I))} \min_{\{\bar{X}^1,\ldots,\bar{X}^n\}} \|\hat{X}^i(I) - \bar{X}^j\|^2.$$
$$(3)$$

This can be done efficiently on GPU — for our problem, K-Means consumes less than 20% of the execution time of the entire forward pass of our method.

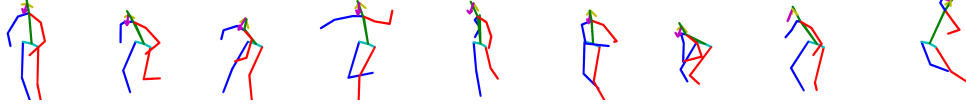

Figure 4: **Example samples from the normalizing flow** $f : X \mapsto z$; $p(z) \sim \mathcal{N}(0, 1)$, trained on a dataset of ground truth 3D SMPL control skeletons $\{X_1, ..., X_N\}$.

**Learning the pose prior with normalizing flows.** In order to obtain $p(X)$, we propose to learn a normalizing flow model in form of the RealNVP network $f$ described in section 3 and the supplementary. RealNVP optimizes the log likelihood $\mathcal{L}_{\text{nf}}(f)$ of training ground truth 3D skeletons $\{X_1, ...X_N\}$ annotated in their corresponding images $\{I_1, ..., I_N\}$ :

$$\mathcal{L}_{\text{nf}}(f) = -\frac{1}{N} \sum_{i=1}^{N} \log p(X_i) = -\frac{1}{N} \sum_{i=1}^{N} \left( \log \mathcal{N}(f(X_i)) - \sum_{l=1}^{L} \log \left| \frac{df_l(X_{li})}{dX_{li}} \right| \right). \quad (4)$$

**2D re-projection loss.** Since the best-of-$M$ loss optimizes a single prediction at a time, often some members of the ensemble $\hat{\mathcal{X}}(I)$ drift away from the manifold of plausible human body shapes, ultimately becoming 'dead' predictions that are never selected as the best hypothesis $m^*$. In order to prevent this, we further utilize a re-projection loss that acts across all hypotheses for a given image. More specifically, we constrain the set of 3D reconstructions to lie on projection rays passing through the 2D input keypoints with the following *hypothesis re-projection loss*:

$$\mathcal{L}_{\text{ri}}(J, G) = \frac{1}{N} \sum_{i=1}^{N} \sum_{m=1}^{M} \left\| Y_i - \pi_{t_i}(\hat{X}^m(I)) \right\|. \quad (5)$$

Note that many of our training images exhibit significant occlusion, so $Y$ may contain invisible or missing points. We handle this by masking $\mathcal{L}_{\text{ri}}$ to prevent these points contributing to the loss.

**SMPL loss.** The final loss terms, introduced by prior work [18, 31, 19], penalize deviations between the predicted and ground truth SMPL parameters. For our method, these are only applied to the best hypothesis $m_i^*$ found above:

$$\mathcal{L}_\theta(G; m^*) = \frac{1}{N} \sum_{i=1}^{N} \|\theta_i - G_{\theta, m_i^*}(I_i)\|; \mathcal{L}_V(G; m^*) = \frac{1}{N} \sum_{i=1}^{N} \|S(\theta_i, \beta_i, \gamma_i) - S(G_{(\theta, \beta, \gamma), m_i^*}(I_i))\| \quad (6)$$

$$\mathcal{L}_\beta(G; m^*) = \frac{1}{N} \sum_{i=1}^{N} \|\beta_i - G_{\beta, m_i^*}(I_i)\|; \mathcal{L}_{\text{rb}}(G; m^*) = \frac{1}{N} \sum_{i=1}^{N} \|Y_i - \pi_{t_i}(\hat{X}^{m_i^*}(I_i))\| \quad (7)$$

Note here we use $\mathcal{L}_{\text{rb}}$ to refer to a 2D re-projection error between the best hypothesis and ground truth 2D points $Y_i$. This differs from the earlier loss $\mathcal{L}_{\text{ri}}$, which is applied across all modes to enforce consistency to the visible *input* points. Note that we could have used eqs. (6) and (7) to select the best hypothesis $m_i^*$, but it would entail an unmanageable memory footprint due to the requirement of SMPL-meshing for every hypothesis before the best-of-$M$ selection.

**Overall loss.** The model is thus trained to minimize:

$$\mathcal{L}(J, G) = \lambda_{\text{ri}} \mathcal{L}_{\text{ri}}(J, G) + \lambda_{\text{best}} \mathcal{L}_{\text{best}}(J, G; m^*) + \lambda_\theta \mathcal{L}_\theta(J, G; m^*) \\ + \lambda_\beta \mathcal{L}_\beta(J, G; m^*) + \lambda_V \mathcal{L}_V(J, G; m^*) + \lambda_{\text{rb}} \mathcal{L}_{\text{rb}}(J, G; m^*) \quad (8)$$

where $m^*$ is given in eq. (1) and $\lambda_{\text{ri}}, \lambda_{\text{best}}, \lambda_\theta, \lambda_\beta, \lambda_V, \lambda_{\text{rb}}$ are weighing factors. We use a consistent set of SMPL loss weights across all experiments $\lambda_{\text{best}} = 25.0$, $\lambda_\theta = 1.0$, $\lambda_\beta = 0.001$, $\lambda_V = 1.0$, and set $\lambda_{\text{ri}} = 1.0$. Since the training of the normalizing flow $f$ is independent of the rest of the model, we train $f$ separately by optimizing $\mathcal{L}_{\text{nf}}$ with the weight of $\lambda_{\text{nf}} = 1.0$. Samples from our trained normalizing flow are shown in fig. 4

## 5 Experiments

In this section we compare our method to several strong baselines. We start by describing the datasets and the baselines, followed by a quantitative and a qualitative evaluation.

Table 1: **Monocular multi-hypothesis human mesh recovery** comparing our approach to two multi-hypothesis baselines (SMPL-CVAE, SMPL-MDN) and state-of-the-art single mode evaluation models [19, 20, 17] on Human3.6m (H36M), its ambiguous version AH36M, on 3DPW and its ambiguous version A3DPW.

| Dataset | Quantization $n$ | 1 | | 5 | | 10 | | 25 | |
|---|---|---|---|---|---|---|---|---|---|
| | Metric | MPJPE | RE | MPJPE | RE | MPJPE | RE | MPJPE | RE |
| H36M | HMR [17] | — | 56.8 | — | — | — | — | — | — |
| | GraphCMR [20] | 71.9 | 50.1 | — | — | — | — | — | — |
| | SPIN [19] | 62.2 | 41.8 | — | — | — | — | — | — |
| | SMPL-MDN | 64.4 | 44.8 | 61.8 | 43.3 | 61.3 | 43.0 | 61.1 | 42.7 |
| | SMPL-CVAE | 70.1 | 46.7 | 68.9 | 46.4 | 68.6 | 46.3 | 68.1 | 46.2 |
| | **Ours** | **61.5** | **41.6** | **59.8** | **42.0** | **59.2** | **42.2** | **58.2** | **42.2** |
| 3DPW | HMR [17] | — | 81.3 | — | — | — | — | — | — |
| | GraphCMR [20] | — | 70.2 | — | — | — | — | — | — |
| | SPIN [19] | 96.9 | **59.3** | — | — | — | — | — | — |
| | SMPL-MDN | 105.8 | 64.7 | 96.9 | 61.2 | 95.9 | 60.7 | 94.9 | 60.1 |
| | SMPL-CVAE | 96.3 | 61.4 | 93.7 | 60.7 | 92.9 | 60.5 | 92.0 | 60.3 |
| | **Ours** | **93.8** | 59.9 | **82.2** | **57.1** | **79.4** | **56.6** | **75.8** | **55.6** |
| AH36M | SMPL-MDN | 113.9 | 74.7 | 98.0 | 70.8 | 95.1 | 69.9 | 91.5 | 69.5 |
| | SMPL-CVAE | 114.5 | 76.5 | 111.5 | 75.7 | 110.6 | 75.4 | 109.7 | 75.1 |
| | **Ours** | **103.6** | **67.8** | **96.4** | **67.1** | **93.5** | **66.0** | **90.0** | **64.2** |
| A3DPW | SMPL-MDN | 159.7 | 82.8 | 154.6 | 83.0 | 149.6 | 80.7 | 122.1 | 76.6 |
| | SMPL-CVAE | 156.6 | 80.2 | 154.5 | 79.9 | 153.9 | 79.8 | 153.1 | 79.8 |
| | **Ours** | **149.6** | **78.5** | **125.6** | **74.4** | **116.7** | **73.7** | **107.8** | **72.1** |

Table 2: **Ablation study on 3DPW** removing either the normalizing flow or the mode re-projection losses and reporting the change in performance.

| Quantization $n$ | | | 5 | | 10 | | 25 | |
|---|---|---|---|---|---|---|---|---|
| Mode reproj. | Flow weight | | MPJPE | RE | MPJPE | RE | MPJPE | RE |
| | | | 86.4 | 57.9 | 84.0 | 57.5 | 79.0 | 56.3 |
| | ✓ | | 84.1 | **57.0** | 81.9 | 56.7 | 77.8 | 55.8 |
| ✓ | | | 82.7 | 57.5 | 79.9 | 57.0 | 76.2 | 55.9 |
| ✓ | ✓ | | **82.2** | 57.1 | **79.4** | **56.6** | **75.8** | **55.6** |

**Datasets and evaluation protocol.** Our evaluation focuses on the Human3.6m (**H36M**) [14, 6] and **3DPW** datasets [45]. H36M is one of the largest datasets of humans annotated with 3D pose using MoCap sensors. As common practice, we train on subjects S1, S5, S6, S7 and S8, and test on S9 and S11. 3DPW is only used for evaluation and, following [20], we evaluate on its test set.

Our evaluation is consistent with [19, 20] - we report two metrics that compare the lifted dense 3D SMPL shape to the ground truth mesh: Mean Per Joint Position Error (**MPJPE**), Reconstruction Error (**RE**). For H36M, all errors are computed using an evaluation scheme known as "Protocol #2". Please refer to supplementary for a detailed explanation of MPJPE and RE.

**Multipose metrics.** MPJPE and RE are traditional metrics that assume a single correct ground truth prediction for a given 2D observation. As mentioned above, such an assumption is rarely correct due to the inherent ambiguity of the monocular 3D shape estimation task. We thus also report MPJPE-

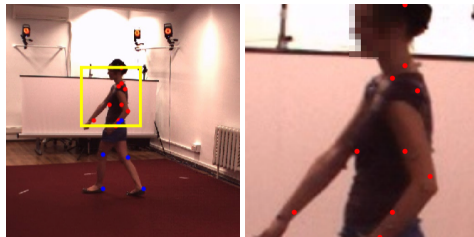

Figure 5: Example image and corresponding annotation from the ambiguous H36M dataset **AH36M**. Best viewed in colour.

$n$/RE-$n$ an extension of MPJPE RE used in [22], that enables an evaluation of $n$ different shape hypotheses. In more detail, to evaluate an algorithm, we allow it to output $n$ possible predictions and, out of this set, we select the one that minimizes the MPJPE/RE metric. We report results for $n \in \{1, 5, 10, 25\}$.

**Ambiguous H36M/3DPW (AH36M/A3DPW).** Since H36M is captured in a controlled environment, it rarely depicts challenging real-world scenarios such as body occlusions that are the main source of ambiguity in the single-view 3D shape estimation problem.

Hence, we construct an adapted version of H36M with synthetically-generated occlusions (fig. 5) by randomly hiding a subset of the 2D keypoints and re-computing an image crop around the remaining visible joints. Please refer to the supplementary for details of the occlusion generation process.

While 3DPW does contain real scenes, for completeness, we also evaluate on a noisy, and thus more challenging version (A3DPW) generated according to the aforementioned strategy.

**Baselines** Our method is compared to two multi-pose prediction baselines. For fairness, both baselines extend the same (state-of-the-art) trunk architecture as we use, and all methods have access to the same training data.

**SMPL-MDN** follows [22] and outputs parameters of a mixture density model over the set of SMPL log-rotation pose parameters. Since a naïve implementation of the MDN model leads to poor performance ($\approx$ 200mm MPJPE-$n = 5$ on H36M), we introduced several improvements that allow optimization of the total loss eq. (8). **SMPL-CVAE**, the second baseline, is a conditional variational autoencoder [40] combined with our trunk network. SMPL-CVAE consists of an encoding network that maps a ground truth SMPL mesh $V$ to a gaussian vector $z$ which is fed together with an encoding of the image to generate a mesh $V'$ such that $V' \approx V$. At test time, we sample $n$ plausible human meshes by drawing $z \sim \mathcal{N}(0, 1)$ to evaluate with MPJPE-$n$/RE-$n$. More details of both SMPL-CVAE and SMPL-MDN have been deferred to the supplementary material.

For completeness, we also compare to three more baselines that tackle the standard single-mesh prediction problem: HMR [17], GraphCMR [31], and SPIN [19], where the latter currently attain state-of-the-art performance on H36M/3DPW. All methods were trained on H36M [14], MPI-INF-3DHP [28], LSP [15], MPII [2] and COCO [23].

## 5.1 Results

Table 1 contains a comprehensive summary of the results on all 3 benchmarks. Our method outperforms the SMPL-CVAE and SMPL-MDN in all metrics on all datasets. For SMPL-CVAE, we found that the encoding network often "cheats" during training by transporting all information about the ground truth, instead of only encoding the modes of ambiguity. The reason for a lower performance of SMPL-MDN is probably the representation of the probability in the space of log-rotations, rather in the space of vertices. Modelling the MDN in the space of model vertices would be more convenient due to being more relevant to the final evaluation metric that aggregates per-vertex errors, however, fitting such high-dimensional (dim=$6890 \times 3$) Gaussian mixture is prohibitively costly.

Furthermore, it is very encouraging to observe that our method is also able to outperform the single-mode baselines [17, 20, 19] on the single mode MPJPE on both H36M and 3DPW. This comes as a surprise since our method has not been optimized for this mode of operation. The difference is more significant for 3DPW which probably happens because 3DPW is not used for training and, hence, the normalizing flow prior acts as an effective filter of predicted outlier poses. Qualitiative results are shown in fig. 6.

**Ablation study.** We further conduct an ablative study on 3DPW that removes components of our method and measures the incurred change in performance. More specifically, we: 1) ablate the hypothesis reprojection loss; 2) set $p(X|I) = \text{Uniform}$ in eq. (3), effectively removing the normalizing flow component and executing unweighted K-Means in $n$-quantized-best-of-$M$. Table 2 demonstrates that removing both contributions decreases performance, validating our design choices.

## 6 Conclusions

In this work, we have explored a seldom visited problem of representing the set of plausible 3D meshes corresponding to a single ambiguous input image of a human. To this end, we have pro-

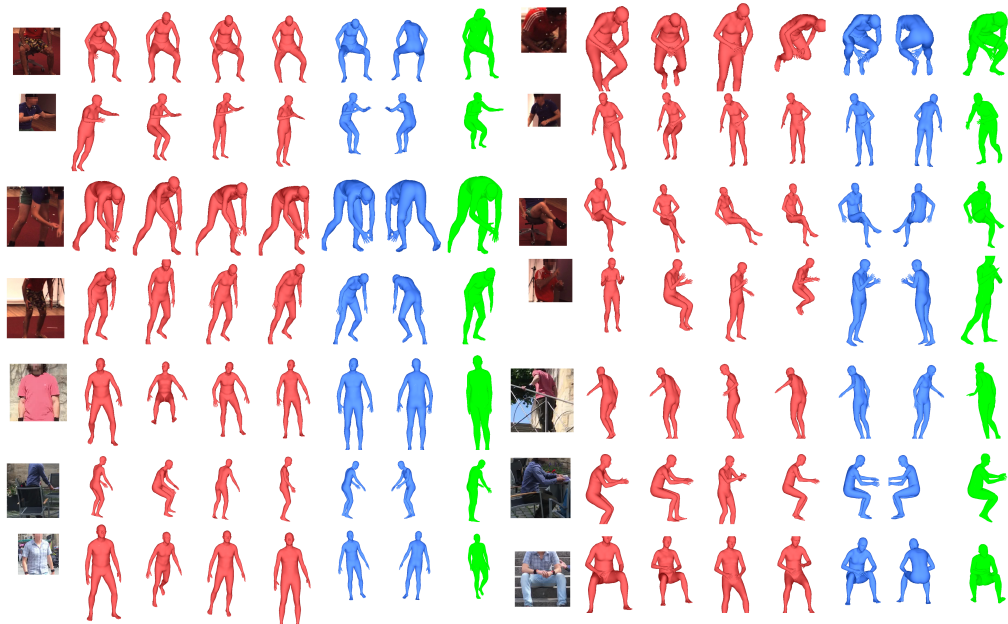

Figure 6: **Qualitative results from** $n = 5$ **quantization on monocular mesh recovery on AH36m and A3DPW.** From left to right, each group of figures depicts the input ambiguous image, five network hypotheses with the closest to the ground truth in blue, and the ground truth pose in green.

posed a novel method that trains a single multi-hypothesis best-of-$M$ model and, using a novel $n$-quantized-best-of-$M$ strategy, allows to sample an arbitrary number $n < M$ of hypotheses.

Importantly, this proposed quantization technique leverages a normalizing flow model, that effectively filters out the predicted hypotheses that are unnatural. Empirical evaluation reveals performance superior to several strong probabilistic baselines on Human36M, its challenging ambiguous version, and on 3DPW. Our method encounters occasional failure cases, such as when tested on individuals with unusual shape (e.g. obese people), since we have very few of these examples in the training set. Tackling such cases would make for interesting and worthwhile future work.

**Acknowledgements**    The authors would like to thank Richard Turner for useful technical discussions relating to normalizing flows, and Philippa Liggins, Thomas Roddick and Nicholas Biggs for proof reading. This work was entirely funded by Facebook AI Research.

## Broader impact

Our method improves the ability of machines to understand human body poses in images and videos. Understanding people automatically may arguably be misused by bad actors. However, importantly, our method is *not* a form of biometric as it does *not* allow the identification of people. Rather, only their overall body shape and pose is reconstructed, but these details are insufficient for unique identification. In particular, individual facial features are not reconstructed at all.

Furthermore, our method is an improvement of existing capabilities, but does not introduce a radical new capability in machine learning. Thus our contribution is unlikely to facilitate misuse of technology which is already available to anyone.

Finally, any potential negative use of a technology should be balanced against positive uses. Understanding body poses has many legitimate applications in VR and AR, medical, assistance to the elderly, assistance to the visual impaired, autonomous driving, human-machine interactions, image and video categorization, platform integrity, etc.

## Footnotes

[2] Theoretically, best-of-$M$ can minimize its loss by quantizing optimally (in the sense of minimum expected distortion) the posterior distribution, which would be desirable for coverage. However, this is *not* the only solution that optimizes the best-of-$M$ training loss, as in the end it is sufficient that *one* hypothesis per training sample is close to the ground truth. In fact, this is exactly what happens; for instance, during training hypotheses in best-of-$M$ are known to easily become degenerate and 'die off', a clear symptom of this problem.

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
