[Supplementary Material]

# 3D Multi-bodies: Fitting Sets of Plausible 3D Human Models to Ambiguous Image Data

## *Supplementary material*

In this section we give a more detailed explanation of the evaluation metrics (appendix A), utilized datasets (appendix B), baseline algorithms (appendix C), training details (appendix D), performance of our method (appendix E) and a more detailed explanation of normalizing flows (appendix F).

## A  Evaluation metrics

More details related to the evaluation metrics, briefly outlined in section 5, are provided in this section.

We report two metrics that compare the lifted dense 3D SMPL shape to the ground truth mesh: Mean Per Joint Position Error (**MPJPE**), Reconstruction Error (**RE**). All errors are computed using an evaluation scheme known as "Protocol #1", as explained below.

For each H36M test skeleton, MPJPE calculates the mean distance between 14 ground truth skeleton 3D joints and the predicted joints obtained by using a fixed linear regressor that maps the array of 6890 3D coordinates of the predicted dense mesh to the skeleton 3D joint coordinates. We report an average of all MPJPE errors measured for each test skeleton. The reconstruction error (RE) is a modification of MPJPE which consists of finding an additional rigid Procrustes alignment between the pair of assessed poses before evaluating the inter-joint distances.

## B  Ambiguous H36m/3DPW

In this section we give a detailed explanation of the generation of the Ambiguous H36m/3DPW datasets (briefly explained in section 5).

We begin with the full size image with a set of 2D joints and apply synthetic occlusions to the subject's body parts by randomly hiding a subset of the 2D keypoints and re-computing a slightly padded image crop around the joints that remained visible. For each image, we randomly choose one of 4 possible strategies for hiding the keypoints: 1) Hiding arm and head keypoints; 2) legs; 3) head; 4) no keypoints hidden[3].

## C  Multi-hypothesis baselines

Here, we describe SMPL-MDN and SMPL-CVAE (section 5) in more detail.

**SMPL-MDN**  SMPL-MDN predicts parameters of a Gaussian mixture defined over the log-rotation parameters $\theta$ of the SMPL kinematic tree. Here, $m$-th Gaussian in the mixture is parametrized with a mean $\mu_m$, covariance matrix $\sigma_m$ and mixture weight $\omega_m$. As noted in section 5, for SMPL-MDN, it was crucial to enable optimization of the total loss (8) in addition to optimizing the log-likelihood of the predicted Gaussian mixture. While the mixture log-likelihood optimizes directly the mixture parameters, the total loss requires a *single* prediction of $\theta$. In order to obtain a single estimate of $\theta$ that can enter the total loss, similar to the Best-of-$M$ loss, we utilize the Gaussian mixture parameters and the ground truth angles $\theta$ to generate a virtual prediction $\hat{\theta}$ that lies

close to the ground truth in the sense of the posterior probability of $\theta$. More specifically, the virtual $\hat{\theta}$ is defined as a weighted combination of mixture means $\mu_m$, where the weights are the posterior probabilities of $\theta$ being assigned to $m$-th mixture component:

$$\hat{\theta} = \sum_{m=1}^{M} \mu_m \frac{p(\theta|\alpha_m, \mu_m, \sigma_m)}{\sum_{n=1}^{M} p(\theta|\alpha_n, \mu_n, \sigma_n)} \; ; \quad p(\theta|\alpha_m, \mu_m, \sigma_m) = \alpha_m N(\theta|\mu_m, \sigma_m), \qquad (9)$$

where $\alpha_m, \sigma_m, \mu_m$ are the weight, variance and mean of the $m$-th mixture component respectively; and $N(\theta|\mu_m, \sigma_m)$ is an evaluation of the multivariate normal distribution with mean $\mu_m$ and variance $\sigma_m$ at $\theta$. Note that the best quantitative results were obtained with fixing $\forall m : \alpha_m = \frac{1}{M}, \sigma_m = 0.001$ and only allowing $\mu_m$ to learn.

This way, the SMPL-MDN regressor $G_{MDN}$ is altered to generate a single prediction $G_{MDN}(I) = (\hat{\theta}, \beta, \gamma, t)$ that enters the total loss (8).

At test-time, following [22], the predicted hypotheses are randomly sampled per-mode predictions $\{(\mu_m, \beta, \gamma, t)\}_{m=1}^{M}$ rather than random samples from the mixture. We observed that randomly sampling from the mixture density gave worse quantitative results.

**SMPL-CVAE** SMPL-CVAE consists of a pair of encoder and decoder networks. The encoder network takes as input the ground truth SMPL mesh $V$ and outputs a Gaussian vector $z$ whose goal is to encode the mode of ambiguity. The decoder $G_{CVAE}(I, z) = (\theta, \beta, \gamma, t)$ then takes as input the $z$, together with the input image $I$, in order to generate the standard tuple of SMPL parameters. At train-time, the network minimizes the total loss (8) and a KL divergence between the predicted distribution of $z$ and a standard multivariate normal distribution $N(0, 1)$.

## D    Training details

Our network is trained in two stages. First we train the original HMR model until convergence according to the training protocol from [20]. We then convert the model to our $n$-quantized-best-of-$M$ architecture and continue training until convergence with an Adam optimizer with an initial learning rate of $10^{-5}$.

## E    Performance analysis

A single inference pass takes on average 0.14s per image on NVIDIA V100 GPU. The overall training time (including the HMR pre-training step) is 5 days on a single gpu.

## F    Normalizing Flows

The idea of normalizing flows is to represent a complex distribution $p(X)$ on a random variable $X$ as a much simpler distribution $p(z)$ on a transformed version $z = f(X)$ of $X$. The transformation $f$ is learned so that $p(z)$ has a fixed shape, usually a Normal $p(z) \sim \mathcal{N}(0, 1)$. Furthermore, $f$ itself must be *invertible* and *smooth*. In this case, the relation between $p(\theta)$ and $p(z)$ is given by a change of variable

$$p(z = f(X)) = \left| \frac{df(X)}{dX} \right| p(X),$$

where, for notational simplicity, we have assumed that $z, X \in \mathbb{R}^D$ are vectors.

The challenge is to learn $f$ from data in a way that maintains its invertibility and smoothness. This is done by decomposing $z = f_L \circ \cdots \circ f_1(X)$ in $n$ layers, where $X_l = f_l(X_{l-1})$, $x = X_n$ and $X = X_0$, and each layer is in turn smooth and invertible. Then one can write

$$\log p(z = f(X)) = \log p(X) + \sum_{l=1}^{L} \log \left| \frac{df_l(X_{l-1})}{dX_{l-1}} \right|.$$

Now the challenge reduces to making sure that individual layers are in fact smooth and invertible and that their inverses and Jacobian determinants are easy to compute. RealNVP [8] does so by writing each layer as $f_l(X_{0:d,l}, X_{d:D,l-1}) = \left( X_{0:d,l-1}, \; X_{d:D,l-1} \odot e^{g_l(X_{0:d,l-1})} + h_i(X_{0:d,l-1}) \right)$ where $g_l, h_l : \mathbb{R}^d \to \mathbb{R}^{D-d}$ are two arbitrary neural networks.

## Footnotes

[3]The selection probabilities are $p(1) = p(2) = p(3) = 0.3, p(4) = 0.1$