[Reviews · NeurIPS 2020]

Review 1

Summary and Contributions: The paper addresses the problem of estimating multiple 3D human bodies that are consistent with a single 2D image. This is important because the 3D pose and shape is ambiguous. Here they explicitly model this ambiguity and produce multiple samples, all of which are consistent with the 2D data. This could be useful for later stages of processing (like tracking).

Strengths: Classical methods for human pose estimation have addressed this problem (some additional references below) but I do not know of a deep learning approach that has done so. This is useful. The numerical results look good and the qualitative results are also good. The method seems to represent valid and sample poses. The paper introduces new losses to make this possible and these are likely to be picked up by others. The multi-hypothesis framework is their key contribution and I think it is a nice contribution to the current field. To make this work, the paper introduces an "n-quantized-best-of-M" loss, which seems useful.

Weaknesses: The authors acknowledge that ambiguous human pose has been considered before (Lines 34-36). They claim to be the first to look at full meshes. This is both a bit narrow and probably not true. Certainly the papers I cite below used meshes, just not learned body models like SMPL. I think these lines should be replaced by a clearer statement of the contribution. Is this the first method to do this in a deep learning framework? The paper does not mention whether code will be made available. Re the pose prior: * It is not clear from the paper what data is used to train the normalizing flow pose prior. * why does fig 4 show samples from the prior as stick figures rather than SMPL bodies? This makes me think the prior is not over SMPL parameters. This is not clear. * in Table 2, it seems that the pose prior makes very little difference. I find it surprising that replacing it with a *uniform* prior works nearly as well. Am I reading this wrong? If a uniform prior works, why bother with the fancy prior? Re the quantitative results: It is surprising that the method is better for M=1 mode. If I am not wrong, this really reduces to HMR with a different pose prior. All the best-of-M stuff shouldn't play a role. Is the method trained like SPIN? In my experience, SPIN is a hard baseline to beat so if you can explain why you beat it for M=1, this will be interesting to your audience.

Correctness: yes, seems correct.

Clarity: Mostly clear. Line 1 (abstract) -- I think posing this work as dealing with "deformable objects" is too broad. This only deals with humans. The loss L_best is defined as the best-of-M loss and then is used in (8) where I think it is actually the new n-quantized-best-of-M loss. This latter loss is never explicitly named. I think you use L_best for both but this is confusing. line 145: ... predicting *a* set...

Relation to Prior Work: I know this is an arXiv paper and I don't know if it is officially published. Regardless, most methods I know of today are using the aggressive cropping method for data augmentation. So this idea is out in the wild and being used. Exemplar Fine-Tuning for 3D Human Pose Fitting Towards In-the-Wild 3D Human Pose Estimation, H Joo, N Neverova, A Vedaldi, https://arxiv.org/abs/2004.03686 It is worth reviewing some of the early history on 3D pose ambiguity and how to represent distributions over pose. Two papers that come to mind are From Sminchescu, this paper looks at the fundamental ambiguities in pose https://hal.inria.fr/inria-00548223/document And this paper from Sidenbladh represented a distribution over 3D poses and exploited this for tracking http://files.is.tue.mpg.de/black/papers/eccv00.pdf In terms of discriminative (learning) methods that also represent ambiguity in poses, there is another work by Sminchisescu on "Discriminative Density Propagation for 3D Human Motion Estimation" http://research.cs.rutgers.edu/~kanaujia/MyPapers/CVPR2005.pdf

Reproducibility: Yes

Additional Feedback: In the text you describe a "hypothesis reprojection loss" and in the results I think this is called a "mode reprojection loss". This change of name is confusing. L 200-201: You say that you can't select best-of-M using the reprojection losses because of the memory footprint of multiple SMPL models. If you are just using the joint locations of SMPL, then you don't need the full mesh. There is no need to generate the full thing to get the joints. So this can be very memory efficient, I think. Rebuttal: I thank the authors for their responses. The rebuttal doesn't actually say how they will clarify the points in the paper. I figure if I was confused, other readers will be too. Please modify the paper to address my comments. On my point above about the prohibitive cost of generating all the vertices (L200-201), I think the authors did not understand my suggestion. I see no reason why the loss function should require all the vertices when the loss is formulated only on the joints. The joint regressor in SMPL is *sparse*. It only depends on a small number of vertices. These vertices can be generated without generating all the other vertices. This is maybe a little-known trick about SMPL -- unless you really need the full mesh, you don't need to generate it. This is certainly true if the loss only uses 2D joints.


Review 2

Summary and Contributions: The paper introduces a method for obtaining multiple plausible solutions for the problem of 3d pose and reconstruction in monocular images. The proposed method targets mostly the images with severe occlusions and partial views, where multiple poses could match the evidence. The method outputs a fixed number of hypotheses which, at testing time, are clustered based on a pose prior learned using normalizing flows.To ensure that all learned hypotheses match the pose in the image, the authors propose to use a 2d reconstruction loss on each hypotheses (not only on the best one). Results are shown on H36m (and a cropped variant of it) and 3DPW datasets.

Strengths: - The paper proposes to tackle a severely occluded/partially visible scenario where many of previous methods either fail or output only a single plausible pose, whereas multiple ones could be plausible. This can have a significant impact in real-world images where this scenario is often encountered.

Weaknesses: The technical contribution of the paper is small. The authors rely on a previous method, HMR, with a slightly modified output. The proposed losses are not new (2d reprojection loss, loss on SMPL parameters, etc). The main contribution is a way to generate and select a plausible set of outputs. However, there are a series of issues that the proposed method raises: - How does the method ensure that the output set of M meshes is diverse and it does not collapse to a single one? (has been addressed in the rebuttal) - Details regarding the learning of p(X|I) are missing, that is, it is unclear how the normalizing flows are conditioned on the input image I. - In the experimental section, the authors only evaluate the *best* out of M solutions. In order to have a better understanding of the capabilities of the method it would have been good to show results for all of them, or, at least also show what is the error for the worst solution. Also, an evaluation of the diversity of the hypotheses pool is missing. (has been addressed in the rebuttal) - What is the value of n (number of clusters) in tables 1 and 2? The visual results (figure 6 for example) suggest that n=3, but is the same value used in the quantitative evaluation? (has been addressed in the rebuttal)

Correctness: The claims and the method seem correct.

Clarity: Yes, the paper is well written and easy to read.

Relation to Prior Work: There are some very recent, relevant works that introduce kinematic priors based on normalizing flows. The authors should include in the final version of the paper a discussion on how their normalizing flow prior relates to these. GHUM & GHUML: Generative 3D Human Shape and Articulated Pose Models (CVPR20) Hongyi Xu, Eduard Gabriel Bazavan, Andrei Zanfir, William T Freeman, Rahul Sukthankar, Cristian Sminchisescu Weakly Supervised 3D Human Pose and Shape Reconstruction with Normalizing Flows (ECCV20) Andrei Zanfir, Eduard Gabriel Bazavan, Hongyi Xu, William T Freeman, Rahul Sukthankar, Cristian Sminchisescu

Reproducibility: No

Additional Feedback:


Review 3

Summary and Contributions: Paper tackles with the problem of 3D pose estimation from heavily occluded images. To solve the problem, a model that predicts multiple poses and a multiple hypotheses loss called "Best-of-M loss" are proposed. A quantization scheme and pose prior are developed to select the n likely poses out of M predictions at test time.

Strengths: - The problem addressed by this paper is quite important and there is not much prior work tackling it. Even state-of-the-art 3D pose estimation methods suffer from heavy occlusions. - Proposed method is neat, simple and works effectively as validated by experiments. - Experiments and ablative analysis are quite strong. Several strong baselines are implemented and analyzed by the authors. - Paper is very well written. It is quite clear and easy to follow.

Weaknesses: - In L104-112 several prior arts are listed. I understand that the task authors tackle is predicting full mesh, but why proposed method is better than [21] or [6]? What makes the proposed approach better than previous methods? From the experiments, the performance difference is clear. However, I am missing the core insights/motivations behind the approach. - In L230, it is indicated that "we allow it (3D pose regressor) to output M possible predictions and, out of this set, we select the one that minimizes the MPJPE/RE metric". Comparison here seems a bit unfair. Instead of using oracle poses, the authors would compute the MPJPE/RE for all of the M or maybe n out of M poses, then report the median error. - It is not clearly indicated whether the curated AH36M dataset is used for training. If so, did other methods eg. HMR, SPIN have access to AH36M data during training for a fair comparison? - There is no promise to release the code and the data. Even though the method is explained clearly, a standard implementation would be quite helpful for the research community. - There is no failure cases/limitations sections. It would be insightful to include such information for researchers who would like to build on this work.

Correctness: - Claims and method are correct according to my evaluation. - Empirical methodology seems correct.

Clarity: - Paper is very well written. - Content is easy to follow. - Mathematical notation is consistent.

Relation to Prior Work: - Contributions and relation to prior art is discussed properly. - It is easy to understand what is proposed differently in comparison to related work. However, there is no insight/motivation statement indicating the advantages/disadvantages of the proposed approach.

Reproducibility: Yes

Additional Feedback: **Post Rebuttal** Thanks authors for the replies to my comments. After reading the rebuttal and comments of other reviewers, I would like to keep my initial score of "7: A good submission; accept".


Review 4

Summary and Contributions: This paper proposed a multi-hypothesis neural network regressor to recover several plausible reconstructions that is compatible with the input data. The proposed regressor is trained with both best-of-M-loss and hypothesis re-projection loss. The flexibility of the regressor is improved by quantizing the best-of-M model by the so-called n-quantized-best-of-M method. Both quantitative and qualitative results prove the effectiveness of the proposed methods.

Strengths: This paper focuses on an interesting problem of reconstructing several dense 3d reconstructions from single and partially occluded views. The proposed method is novel and technically practicable. The n-quantized-best-of-M method is well designed which makes the best-of-M model more flexible to other applications. The experimental results prove the effectiveness of the proposed methods.

Weaknesses: This paper assumes that the 2D locations of the body joints are known for all input images and utilize a re-projection loss to constraint all hypothesis. However, to accurately capture 2D joint location in partially occluded views is also a challenging task which limits the flexibility of the proposed method. More implementation details should be provided to improve the reproducibility of the proposed method, e.g. what kind of training images are utilized as input for H36M and AH36M datasets, the batch size, weight decay and number of epochs etc.

Correctness: The proposed method is technically correct.

Clarity: The paper is clearly written, but lots of implementation details should be added.

Relation to Prior Work: This paper has already presented the difference in the section 2.

Reproducibility: No

Additional Feedback: In L74, 'Human36M' should be Human3.6m. In L204, the value of λ_{rb} is missing In Table 1, have the methods been retrained on AH36M? The results of SPIN are slightly different from the original paper, 41.8 in Table 1 and 41.1 in [18]. It is better to show some qualitative results in the ablation study after removing each component. Are the results in Figure 6 ranked by the generated weight? Final decision: The rebuttal has explained most of the questions. I will keep the original rating.

[Author Response · NeurIPS 2020]

We would like to thank our reviewers for their constructive comments.

**R1: Not first to look at full meshes. Cite Sminchescu et al., ... Soften claims?** Thank you for the references, which
we will add. Accordingly, we will limit our claim to deep learning approaches for human pose reconstruction. **R1, R3:**
**Release code and data.** We will release those, along with pre-trained models. **R1: Not clear... what data is used**
**to train the norm. flow. Why does fig 4 show stick figures?** The prior is on the SMPL *pose* and *shape* parameters,
which we reparameterize as 24 3D control joints (L185-187). This was shown empirically to produce better results.
**R1: Pose prior makes little difference in Tab. 2.** Yes, the benefit of the n.f. prior is small in 3DPW primarily
because there are limited occlusions/ambiguities in this dataset. In this case, the benefit of the re-projection loss (Eq
5) dominates (when the latter is removed, the effect of the prior is more pronounced - rows 1, 2 in Tab. 2). **R1: The**
**improvements for $M = 1$ mode are surprising. Why?** Best-of-M method should not *per-se* benefit the $n = 1$
case, but there are other differences: the $n = 1$ hypothesis is obtained by quantizing from the $M = 100$ model while
using the pose prior to re-weight the predicted poses, which is the potential source of improvement (L264-266). **R1:**
**L200-201: If only joint locations of SMPL used, $L_V$ can be computed for all hypotheses.** Actually, the loss $L_V$
on L200-201 is based on *the full set of 6890 SMPL vertices*, which is why it is so expensive (L199-201).

**R2: HMR and SMPL losses already published. The main contribution is a way to generate and select a plausible**
**set of outputs.** Yes, our contribution is *n-quantized-best-of-M* — a better method for multi-hypothesis generation.
In agreement with the other reviewers, we believe this to be a non-trivial step forward. Also, while the reprojection
loss is not new, the way it is applied to all hypotheses, best and not (L53-59, L191, Eq 5), is new in best-of-M and
brings a sizeable improvement for this class of methods (Tab 2 rows 1-2 vs 3-4). **R2: How does the method ensure**
**diversity?** This is achieved automatically, by the best-of-M loss: 'guessing' only a single hypothesis would be
disadvantageous compared to outputting M diverse guesses as that has a higher chance that one would have low
error. Still, previous best-of-M implementations fail to reliably converge to this optimal solution, resulting in some
modes to degenerate. An advantage of our loss (Eq 5) is that it helps to keep all modes active (L39-53). **R2:**
**How is the n.f. conditioned on the image?** Please see L171-183. **R2: Evaluation of all hypotheses/diversity.**
We follow the standard evaluation style in this area [21] (L227-228) adapted for our task of 3D mesh prediction.
Achieving a low mean/median error of all hypotheses would in fact show that the diversity of the solution is low,
which defeats the purpose of spanning the space of plausible hypotheses. However, we agree that evaluating the
diversity itself is valuable and we present results in Tab. I: on average, our method yields the most diverse predictions.

**R2: Number of clusters in Tables 1 and 2?** We apologise for a typo on
L231, which should read $n \in \{1, 5, 10, 25\}$. The value of $M$ is 100. In
Figure 6, we show the best hypothesis (in blue) and 3 other hypotheses
from the $n = 5$ quantization. We omitted the last hypothesis to allow space
for 2 columns of results over 2 datasets. We agree that this is confusing and
we will include the missing hypothesis for a future version of the paper.

| Num Modes | 5 | 10 | 25 |
|---|---|---|---|
| SMPL-MDN | 47.3 | 47.4 | 49.7 |
| SMPL-CVAE | 2.1 | 2.4 | 2.5 |
| **Ours** | 45.8 | 53.8 | 58.3 |

Table I: **Hypothesis diversity on AH36m**. For a pair of meshes, diversity is an average 3D distance between corresponding control joints. We first compute a mean diversity over all pairs of hypotheses in an image and report an average over the test set. Tab. 1 extension.

**R3: Core insight?** Best-of-M models have been shown to outperform
alternatives (such as VAE/MDN) in tasks with constrained output spaces
(e.g. 2D keypoints [Rupprecht et al. ICCV 2017], or our 3D control joints
are less complex than, say, the space of natural images). We thus selected
best-of-M as the core of our contribution, while providing improvements
that make best-of-M feasible/better for deformable 3D shapes. We will
expand the paper with this motivation. **R3: Best hypothesis evaluation a**
**bit unfair.** Reporting the best prediction is standard in best-of-M models
[21] (and comes with the definition). This is because achieving a low mean/median error would in fact show that the
diversity of the solution is low, which defeats the purpose of spanning the space of plausible hypotheses.

**R3, R4: AH36M dataset is used for training? What about HMR and SPIN?** The AH36M dataset is used for
training *all* compared methods (we will clarify this in the paper). **R3: Failure cases?** Failure cases include testing
on busy crowd scenes which include multiple people inside a single bounding box, or individuals of unusual shape
(e.g. obese people), since we have very few of these examples in the train set. We will add a remark to the paper.

**R4: Accurately capturing 2D joint locations in occluded views is challenging.** We mask the loss with a visibility
flag for missing keypoints. This allows to learn from images where annotators could not identify all keypoints. We
will clarify this in a future version of the paper. **R4: More details on training and data** We will release code, data,
pretrained models and add these specific details to the sup. mat. **R4: The results of SPIN are slightly different from**
**the original paper, 41.8 in Table 1 and 41.1 in [18].** We take the results from the pretrained models released by the
authors which differ slightly from the paper. **R4: Qualitative results in the ablation study.** We agree this would be
a useful addition, and will include this in the sup. mat. **R4: Are the results in Figure 6 ranked by the generated**
**weight?** No, but we agree this would improve this figure and we will alter the order in a future version of the paper.

[Meta-Review · NeurIPS 2020]

All reviewers recommend acceptance. Methodology is interesting, but some of the claims in the paper are questionable and multiple references are missing. In the final version, please include the references indicated by R1 and R2 as well as the clarifications requested during reviewing. In particular please correctly relate to prior normalizing flow work to build kinematic priors, published at CVPR 2020 and ECCV2020, and currently not cited, as indicated by R2.